# Interlaboratory validation trial report on multiplex real-time PCR method for molecular serotyping and identification of the 30 major clonal complexes of *Listeria monocytogenes* circulating in food in Europe

Karine Capitaine,[1] Sandrine Te,[1] Adrien Asséré,[1] Hana Plodková,[2] Valerie Michel,[3] Pauline Sabrou,[3] Erwan Bourdonnais,[4] Guillaume Gillot,[5] Nassim Mouhali,[5] Thomas Brauge,[6] Cécile Dumaire,[6] Carole Feurer,[7] Baptiste Houry,[7] Stefanie Lueth,[8] Zsuzsanna Sréterné Lancz,[9] Gabriella Centorotola,[10] Fabrizia Guidi,[10] Marina Torresi,[10] Tone Mathisen Fagereng,[11] Taran Skjerdal,[11] Hugo Guedes,[12] Gonçalo Nieto Almeida,[12] Laurentiu Mihai Ciupescu,[13] Paula Ågren,[14] Monica Ricão,[14] Elisabet Marti,[15] Wilma Jacobs-Reitsma,[16] Angela van Hoek,[16] Benjamin Félix[1]

**ABSTRACT** The performance of a new method developed in 2021 by the European Union Reference Laboratory (EURL) for *Listeria monocytogenes* based on 12 multiplex real-time PCR, allowing the identification of the molecular serotype and the 30 major *L. monocytogenes* multilocus sequence typing clonal complexes (CC), was assessed through a European interlaboratory validation trial (ILVT). This ILVT was adapted from ISO standard 16140 part 6. Overall, 98 blinded pure strains of *Listeria* (*monocytogenes* or spp.), previously characterized by the EURL, were sent to 15 laboratories distributed in 11 countries. The molecular serotype had to be identified for 20 strains of the ILVT panel, while CC identification had to be performed for the whole panel. The results of the 12 multiplex real-time PCR were reproducible between the participating laboratories with high individual concordance values for molecular serotyping (100%) and CC identification (90.8%–100%) irrespective of DNA extraction protocols, PCR master mixes, and thermocycler diversity. Master mixes identified as incompatible with some of the multiplex real-time PCR were excluded from the method. The overall concordance of the results was sufficient for the method to be confidently applied in other laboratories involved in *L. monocytogenes* typing.

**IMPORTANCE** This interlaboratory validation trial, coordinated by the European Union Reference Laboratory for *Listeria monocytogenes*, was the final step to assess the performance of the multiplex real-time PCR method developed and published by B. Félix, K. Capitaine, S. Te, A. Felten, et al. (Microbiol Spectr 11:e0395422, 2023, https://doi.org/10.1128/spectrum.03954-22). Different combinations of parameter settings were applied in 15 French and European laboratories involved in *L. monocytogenes* typing. It was a prerequisite to establish this new real-time PCR method as a standard for rapid molecular serotyping and clonal complex identification. The accuracy and reproducibility of the results obtained on the panel of 98 strains of *L. monocytogenes* sent to the participants proved that the real-time PCR was suitable for use in their conditions. Rapid screening of strains is therefore now possible, and the method provides a valuable tool for epidemiological investigations to identify food-associated strains during listeriosis outbreaks.

**KEYWORDS** *Listeria monocytogenes*, real-time PCR, clonal complex, MLST, molecular serotype, interlaboratory validation trial, Europe

**Peer Reviewer** Giovanna Franciosa, Istituto Superiore di Sanità, Rome, Italy

Address correspondence to Karine Capitaine, karine.capitaine@anses.fr.

The authors declare no conflict of interest.

See the funding table on p. 10.

*L*isteria monocytogenes is a foodborne pathogenic bacterium responsible for listeriosis, a zoonotic disease. It contaminates the food chain from its primary reservoirs—animal and soil—colonizes the food production environment, and is then transmitted to humans through the consumption of contaminated food products. Although rare, invasive listeriosis is of public health concern because of its severity associated with a high lethality rate reported in Europe at 19.7% in 2023 and its potential to cause outbreaks (1–3). Overall, the European Union trend for human cases showed a significant and gradual increase between 2019 and 2023 (4). Current bacteriological surveillance and investigation are commonly conducted using core genome multilocus sequence typing (cgMLST) (5), chosen as the reference method for *L. monocytogenes* molecular typing, because of its reproducibility and ultimate discriminatory power (5). Despite its outstanding potential, cgMLST requires the whole-genome sequencing (WGS) of the strain, which can be time-consuming, labor-intensive, and cost-effective. Therefore, molecular serotyping (6, 7) is generally used as a pre-screening method prior to WGS and cgMLST analysis, reducing the number of strains to sequence. However, this method creates solely four molecular groups within *L. monocytogenes* species (IIa, IIb, IIc, and IVb), which is few considering the large diversity of strains circulating in food production.

The multilocus sequence typing (MLST) classifies *L. monocytogenes* into clonal complexes (CCs) and sequence types (STs), which are systematically used to describe its population structure (8). STs are defined as the unique association of alleles from seven housekeeping genes, and a CC is described as a cluster of STs sharing at least six alleles (9). CCs descend from a common ancestor and have accumulated differences predominantly through mutations. CCs evolve slowly over large temporal and geographic scales (10–12). To date, 266 CCs have been identified in the MLST online database (https://bigsdb.pasteur.fr). Only 30 CCs account for the majority of the strains circulating in the food production chain in Europe (13).

The European Union Reference Laboratory (EURL) for *L. monocytogenes* is coordinating a network of 41 National Reference Laboratories (NRLs) involved in strain typing in their country. NRLs are responsible for epidemiological investigation in the context of outbreaks and are demanding a harmonized, rapid, and discriminative method for strain screening.

A recently developed method makes it possible to identify 30 MLST CCs using real-time PCR through 1 duplex and 11 triplex assays. This method divides the molecular serotypes IIa into 18 CCs, IIb into 6 CCs, IVb into 5 CCs, and IIc into 1 CC (13). CC has become the common language for *L. monocytogenes* typing, pivotal information for outbreak definition, risk assessment, and virulence for humans (14, 15). An attribute of this typing method is its ease of use by different laboratories, thereby leading to a standard method and typing nomenclature. In the present study, results of an interlaboratory validation trial (ILVT) of the 30 CC identification multiplex real-time PCR scheme developed by Félix et al. (13) are presented. This ILVT, coordinated by the EURL for *L. monocytogenes*, is a prerequisite to establish this method as a standard for rapid CC identification. This ILVT also includes the validation of two triplex real-time PCR for molecular serotyping identification according to Vitullo et al. (7).

## MATERIALS AND METHODS

### Panel of strains

Sets of the same 98 coded strains, epidemiologically unrelated, were divided into four panels called α, β, λ, and Δ (Fig. 1) and analyzed individually by the participants. The strains, stored in stab tubes (Stock Culture Agar, Bio-rad, Marnes-la-Coquette, France), were distributed to 10 NRLs for *L. monocytogenes* in Europe, 4 French agro-industrial technical institutes, and 1 ANSES laboratory, all involved in *L. monocytogenes* national surveillance (Fig. 2). Strains were mainly selected from former European projects involving the EURL for *L. monocytogenes* and from the strains routinely received at ANSES over the past 13 years. A large majority of them had been isolated from various food

products or food processing environments; one was a clinical isolate from a human case of listeriosis (16). Ninety strains belonged to 1 of the 30 CCs identified by the method. Three strains were selected within each detected CC. For eight strains, the CC could not be detected by the method. Four of them were *L. monocytogenes,* and four were *L.* spp.: *Listeria innocua, Listeria ivanovii, Listeria seeligeri,* and *Listeria welshimeri*. These *L.* spp. non-*monocytogenes* strains were selected because they are known to be frequent food chain contaminants and can potentially be isolated along with *L. monocytogenes*.

Among the 98 coded strains, 11 were requested by each participant to be used for the molecular serotyping assessment and belonged to Δ panel (Fig. 1). Among the 11 strains, 2 belonged to IIa, 1 to IIb, 3 to IIc, 1 to IVb, and the 4 *L.* spp. non-*monocytogenes* above cited (Table 1). Among the remaining strains, nine were chosen by the participants on the basis of their CC to test three strains per molecular serotype: IIa, IIb, and IVb. Overall, 20 strains were used for molecular serotyping assessment.

Strains were first identified by the coordinating laboratory by WGS, standard for MLST (9), and molecular serotyping (17). The original CC of the coded strains and their molecular serotype were revealed only when results of the multiplex real-time PCR assay from all participants had been reported to the coordinator of the study.

## Multiplex real-time PCR

The participants had to follow the multiplex real-time PCR parameters as previously described (13), with two major changes: PCR CC14-ST91 was removed, and PCR CC18 was moved to the triplex PCR CC8-CC18-CC121. The updated method was made public via the Zenodo platform (18, 19). The participants were encouraged to use primer and probe sets provided by the coordinating laboratory. The sets of freeze-dried primers and probes were produced from the same batch by the supplier (TIB Molbiol, Berlin, Germany; https://www.tib-molbiol.de). Two different positive controls were provided: a plasmid mixture positive with all the PCRs, and a pure strain DNA extract for specific interpretation of PCR CC101.

## Real-time PCR results and statistical analysis

Each participant separately interpreted the real-time PCR amplification results, according to the coordinator's instructions. The results obtained by the participants for each of the

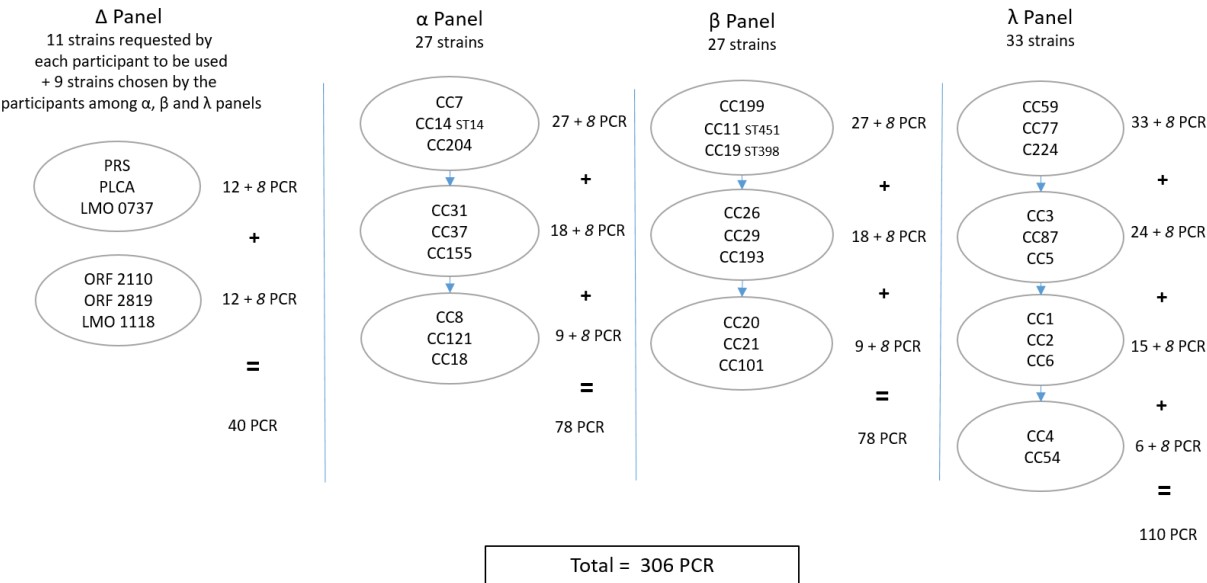

FIG 1   Distribution of the 98 strains of the Inter-laboratory validation trial into four panels. Each circle represents one multiplex real-time PCR and its targeted CC. Beside circle is reported the number of PCR performed by the participants. "+8"in italics relies on four *Listeria* non-*monocytogenes* and four *Listeria monocytogenes* strains with CC not identified by the method.

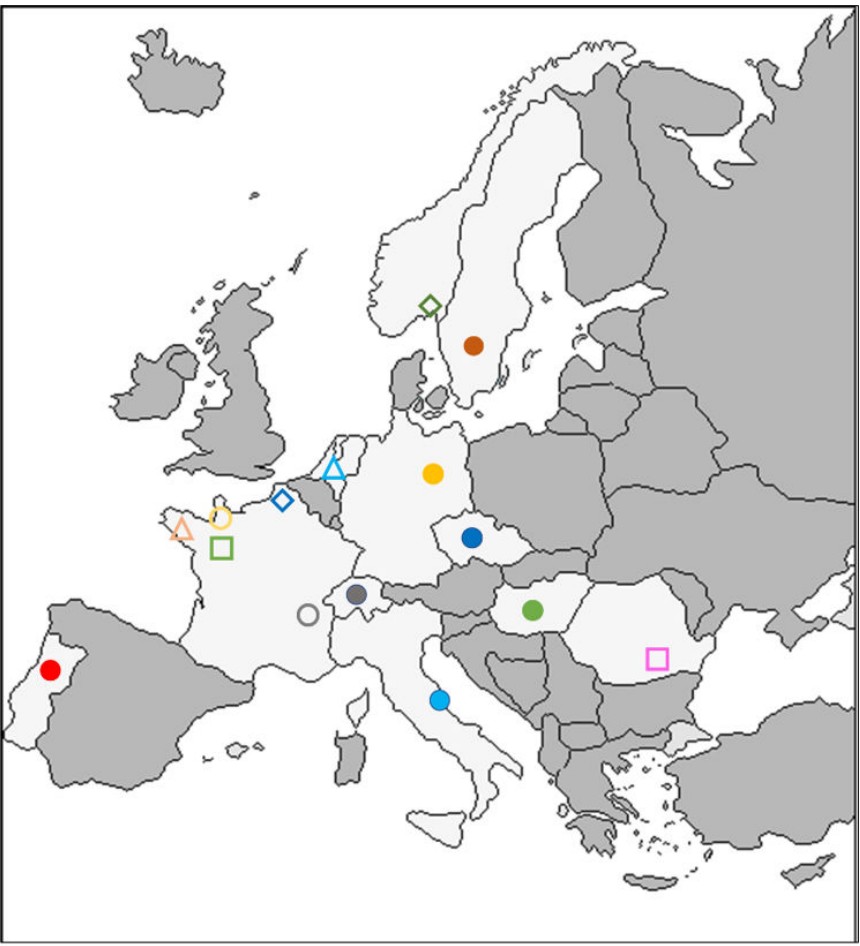

**FIG 2** Map of the 15 participating laboratories.

98 strains were set up according to the positive controls. Cycle threshold (Ct) values for each positive real-time PCR, CC identification, and molecular serotype of the strains were tabulated and sent to the coordinating laboratory. Results from the 15 laboratories were compared with the coordinating laboratory's results. Statistical analysis was performed for each molecular serotype and targeted CC, with calculations of sensitivity, specificity, and accuracy. Sensitivity shows true positive (TP) on false-negative (FN) rate, specificity shows true negative (TN) on false-positive (FP) rate, and accuracy shows both sensitivity and specificity rates combined. TP designates positive real-time PCR identification if truly present, while TN designates real-time PCR identification not identified if truly absent. Positive controls were not included in the concordance calculation.

$$\text{Sensitivity} = \frac{\text{TP}}{(\text{TP} + \text{FN})}, \quad \text{Specificity} = \frac{\text{TP}}{(\text{TP} + \text{FN})}, \quad \text{Accuracy} = \frac{\text{TP} + \text{TN}}{(\text{TP} + \text{TN} + \text{FP} + \text{FN})}.$$

## RESULTS

### Real-time PCR conditions

For real-time PCR, 9 different DNA extraction methods, 10 different master mixes, and 6 different thermocyclers were used by the participants (Table S1). Overall, 14 different combinations of three experimental settings were implemented by the participants.

**TABLE 1** Molecular serotyping PCR assessment panel.

| Molecular serotype | Number of different strains tested | Strains MLST + *Listeria* species diversity | |
|---|---|---|---|
| | | Strains requested to participants to be used | Strains chosen by participants among the ILVT panel based on their CC |
| IIa | 24 | ST207, ST412 | ST8, ST14, ST18, ST21, ST26, ST31, ST37, ST121, ST155, ST193, ST199, ST204, ST206, ST511, ST691 |
| IIb | 18 | ST392 | ST3, ST5, ST59, ST77, ST87, ST224 |
| IIc | 3 | ST9, ST622 | _[a] |
| IVb | 15 | ST389 | ST1, ST2, ST4, ST6, ST54, ST179, ST308 |
| L | 4 | *L. innocua, L. ivanovii, L. seeligeri, L. welshimeri* | – |

[a]"–", No strain of this molecular serotype.

## Molecular serotype identification

All 15 participants correctly identified the molecular serotype of the 20 strains. For the 11 strains commonly typed by the participants, all laboratories successfully assigned the 4 *L.* spp. non-*monocytogenes* strains to the molecular serotype L, and the 7 *L. monocytogenes* strains to the correct molecular serotypes IIa ($n = 2$, CC207 and CC412), IIb ($n = 1$, CC392), IIc ($n = 3$, CC9), and IVb ($n = 1$, CC389). For the nine strains chosen by the participants, all laboratories successfully assigned the correct molecular serotype for the 24 IIa, 18 IIb, and 15 IVb strains. Overall, the strains encompassed 27 different CCs and 34 different STs (Table 1).

## Clonal complex identification

### Positive identification results

Of the 15 participants, 10 assigned all of the 98 strains to the expected CC (Table 2). The overall concordance was 98.9% (1,335/1,350). The individual concordance for those labs that failed to identify expected CCs was 90.8% (92/98) for lab 6, 96.9% for labs 1 and 3, 98.0% for lab 14, and 99% for lab 7. For concordant results, the Ct obtained for the PCR was between 10 and 30 (Fig. 3). The Ct variability observed was related to the range in DNA concentrations and PCR conditions (Table S1).

Among the 10 labs that obtained a full concordance, 6 laboratories achieved it on the first attempt, while the other labs required a second attempt (Table 2). These latter labs failed due to Ct values > 30 for the PCR on strains or on the positive control. At the second try, labs 8 and 10 changed the master mix to solve the failed PCR (Table S1), lab 11 used its own primers and probes from a local supplier, most likely more compatible with its master mix, and finally, lab 15 re-performed its PCR successfully without technical change.

For lab 7, the second attempt partially solved the failures. They were related to contamination of two strains leading to multiple identifications. After re-extraction, one strain gave concordant results, while the other remained contaminated. This issue might be related to multiple contaminations from the stab tube.

For four laboratories, the second attempt did not solve the failed PCR. For labs 1 and 3, failures were related to the late detection of the positive control for PCR CC193. For lab 13, PCR amplification did not work for two strains: one CC20 and one CC37. For lab 6, failures were related to late detection of the positive control for CC3 and CC20, and late detection of two CC3 strains. For positive control failures, an advanced optimization is required to figure out possible reagent incompatibility. For single-strain failures, issues with DNA extraction, quantification, or dilution could explain the repeated failure.

For seven laboratories, a positive amplification was observed with (i) the *L. innocua* strain for labs 1, 2, 3, 6, 7, and 11 with PCR CC101 and (ii) the *L. welshimeri* strain for lab 8 with PCR CC21 (Table 3). For these strains, PCR PLCA of molecular serotyping (*L. monocytogenes* species identification) was negative, and they were identified as *L.* non-*monocytogenes* by the participants.

**TABLE 2** Results of the multiplex PCR assay of *Listeria* strains performed by 15 participating laboratories[a]

| Species | Molecular serotype | CC | No. of strains | Laboratories | | | | | | | | | | | | | | |
|---|---|---|---|---|---|---|---|---|---|---|---|---|---|---|---|---|---|---|
| | | | | 1 | 2 | 3 | 4 | 5 | 6 | 7 | 8 | 9 | 10 | 11 | 12 | 13 | 14 | 15 |
| *Listeria monocytogenes* | IIa | CC7, CC8, CC11-ST451, CC14-ST14, CC18, CC19-ST398, CC20, CC21, CC26, CC29, CC31, CC37, CC101, CC121, CC155, CC193, CC199, CC204 | 54 | 51(3) | 54 | 51(3) | 54 | 54 | 52(6) | 53(2) | 54(19) | 54 | 54(18) | 54(12) | 54 | 52(2) | 54 | 54(3) |
| | IIb | CC3, CC5, CC59, CC77, CC87, CC224 | 18 | 18 | 18 | 18 | 18 | 18 | 15 | 18 | 18(7) | 18 | 18(3) | 18(3) | 18 | 18 | 18 | 18 |
| | IIc | CC9 | 3 | 3 | 3 | 3 | 3 | 3 | 3 | 3 | 3 | 3 | 3 | 3 | 3 | 3 | 3 | 3 |
| | IVb | CC1, CC2, CC4, CC6, CC54 | 15 | 15 | 15 | 15 | 15 | 15 | 15 | 15 | 15(15) | 15 | 15 | 15 | 15 | 15 | 15 | 15 |
| *Listeria monocytogenes* others CC | IIa | CC207, CC412 | 2 | 2 | 2 | 2 | 2 | 2 | 2 | 2 | 2 | 2 | 2 | 2 | 2 | 2 | 2 | 2 |
| | IIb | CC392 | 1 | 1 | 1 | 1 | 1 | 1 | 1 | 1 | 1 | 1 | 1 | 1(1) | 1 | 1 | 1 | 1 |
| | IVb | CC389 | 1 | 1 | 1 | 1 | 1 | 1 | 1 | 1 | 1 | 1 | 1 | 1 | 1 | 1 | 1 | 1 |
| *Listeria* spp. | L | *L. innocua, L. welshimeri, L. seeligeri, L. ivanovii* | 4 | 4 | 4 | 4 | 4 | 4 | 4 | 4 | 4 | 4 | 4 | 4 | 4 | 4 | 4 | 4 |
| First assay success (%) | | | | 96.9 | 100 | 96.9 | 100 | 100 | 90.8 | 98.0 | 53.1 | 100 | 78.6 | 83.7 | 100 | 98.0 | 100 | 96.9 |
| Total concordance (%) | | | | 96.9 | 100 | 96.9 | 100 | 100 | 90.8 | 99.0 | 100 | 100 | 100 | 100 | 100 | 98.0 | 100 | 100 |

[a]Results are the final number of strains, after the first and second attempts, identified with the expected PCR CC. The number of strains requiring a second PCR assay for interpretation is presented in brackets.

## Negative identification results

The strain analysis was performed using multiplex PCR (1 duplex and 11 triplex). When one PCR was positive, the other had to be negative. The overall concordance for these negative results was 99.9% (10,714/10,724).

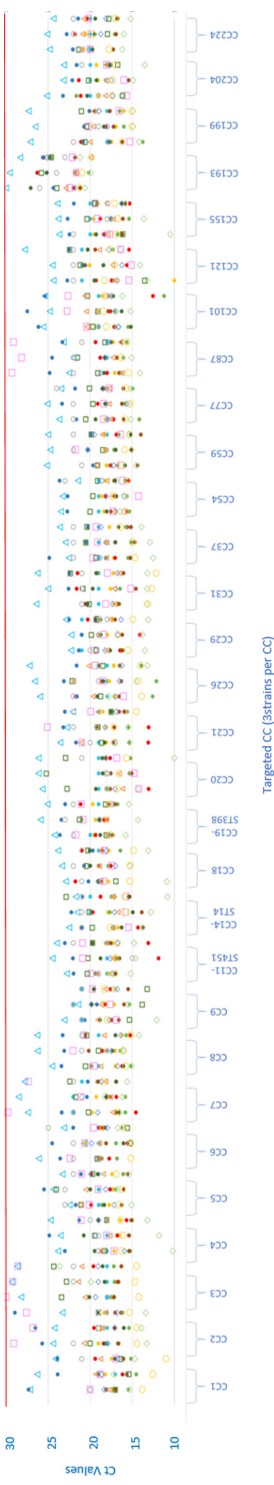

**FIG 3** CC PCR cycle threshold values for each participating laboratory. The red line shows the PCR positivity limit. Coloured items depend on lab designation on the map (Figure 2).

## Statistical analysis

Statistical analysis was performed to evaluate molecular serotyping and CC identification results (Table 3).

For molecular serotyping, the statistical review of the results was 100% for specificity, sensitivity, and accuracy (Table 3).

For CC identification, the statistical review of the results indicated a high degree of specificity, sensitivity, and accuracy for the 30 real-time PCR, with 100% for 22 of them. For CC14-ST14, CC18, CC21, and CC101, specificity results were 99.5%, 99.8%, 99.5%, and 97.1%, respectively, due to one contaminated sample for the first two and positive amplifications of *L. innocua* and *L. welshimeri* for the last two. For CC3, CC20, CC37, and CC193, the sensitivity results were 93.3%, 91.1%, 97.8%, and 86.7%, respectively, due to CC identification failures caused by positive control amplification or PCR CC amplification failures. For CC3, CC14-ST14, CC18, CC20, CC21, CC37, CC101, and CC193, accuracy was between 97.6% and 99.8%.

## DISCUSSION AND PERSPECTIVES

This ILVT involved 15 laboratories and evaluated two real-time triplex PCR assays for the identification of *L. monocytogenes* molecular serotyping according to Vitullo et al. (7) and one duplex and nine triplex assays for the identification of 30 CCs. The failures encountered by the participants were related to reagent incompatibility and contamination, which were solved through real-time PCR optimization. For one *L. innocua* and one *L. welshimeri*, false-positive amplifications were found with two different real-time PCR CC identification, without any consequence, as both strains were identified as non-*monocytogenes* during molecular serotyping identification. Overall, the ILVT results proved the reliability of the method to identify the genus *Listeria*, the species *monocytogenes*, all molecular serotypes, and the 30 targeted CCs.

Previously, two large PCR ILVT studies were performed on *L. monocytogenes* molecular serotyping. One was conducted in 2005, using six PCR assays, on 90 strains, including five laboratories. This ILVT was performed without reagent standardization. The second was conducted in 2023, using seven real-time PCR assays, on 46 strains, including 16 laboratories. It was conducted with a highly standardized procedure, using the same real-time PCR thermocycler and a single bulk preparation of each reagent, aliquoted and dispatched to the participants. In the present study, standardization of real-time PCR parameters was impossible considering the diversity of the participating laboratories, for which accessibility to the reagent can rely on a local supplier and the real-time PCR thermocycler in place. Despite this, the results of molecular serotyping of our study were higher in specificity, sensitivity, and accuracy than those obtained by Pamboukian et al. (20), without high standardization of DNA extraction method and real-time PCR conditions, underlining the versatility of the real-time PCR scheme developed.

In 2013, a multicenter trial was conduced on another pathogen, involving eight laboratories for the identification of toxins of Clostridium botulinum. 81 DNA extracted were provided (21). DNA extracts were provided, and the thermocycler model used for PCR was also standardized, limiting the possibility of variation in results between laboratories. Despite the additional variation introduced by the variability in DNA extraction method, the results obtained for CC identification of our study are comparable to those of the study cited above.

In comparison to molecular serotyping, CC identification provides a five times more discriminative method for strain typing. The multiplex real-time PCR assays are suitable for rapid screening of strains of *L. monocytogenes* and are valuable for epidemiological investigations to identify food-associated strains during listeriosis outbreaks. The rapid screening also provides the capacity for rapid and high-throughput strain identification, in the context of large strain screening or diversity studies, for instance.

**TABLE 3** Statistical analysis of the identification results

| Real-time PCR assay | n | Specificity (%) | Sensitivity (%) | Accuracy (%) |
|---|---|---|---|---|
| IIa | 300 | 100 | 100 | 100 |
| IIb | 300 | 100 | 100 | 100 |
| IIc | 300 | 100 | 100 | 100 |
| IVb | 300 | 100 | 100 | 100 |
| L | 300 | 100 | 100 | 100 |
| CC1 | 345 | 100 | 100 | 100 |
| CC2 | 345 | 100 | 100 | 100 |
| CC3 | 480 | 100 | 93.3 | 99.4 |
| CC4 | 210 | 100 | 100 | 100 |
| CC5 | 480 | 100 | 100 | 100 |
| CC6 | 345 | 100 | 100 | 100 |
| CC7 | 525 | 100 | 100 | 100 |
| CC8 | 255 | 100 | 100 | 100 |
| CC9 | 300 | 100 | 100 | 100 |
| CC11-ST451 | 525 | 100 | 100 | 100 |
| CC14-ST14-206-399 | 525 | 99.8 | 100 | 99.8 |
| CC18 | 255 | 99.5 | 100 | 99.6 |
| CC19-ST398 | 525 | 100 | 100 | 100 |
| CC20 | 255 | 100 | 91.1 | 98.4 |
| CC21 | 255 | 99.5 | 100 | 99.6 |
| CC26 | 390 | 100 | 100 | 100 |
| CC29 | 390 | 100 | 100 | 100 |
| CC31 | 390 | 100 | 100 | 100 |
| CC37 | 390 | 100 | 97.8 | 99.7 |
| CC54 | 210 | 100 | 100 | 100 |
| CC59 | 615 | 100 | 100 | 100 |
| CC77 | 615 | 100 | 100 | 100 |
| CC87 | 480 | 100 | 100 | 100 |
| CC101 | 255 | 97.1 | 100 | 97.6 |
| CC121 | 255 | 100 | 100 | 100 |
| CC155 | 390 | 100 | 100 | 100 |
| CC193 | 390 | 100 | 86.7 | 98.5 |
| CC199 | 525 | 100 | 100 | 100 |
| CC204 | 525 | 100 | 100 | 100 |
| CC224 | 615 | 100 | 100 | 100 |
| Total | 13,560 | 99.9 | 99 | 99.8 |

A short-term useful application of this assay is its use in DNA extraction from *L. monocytogenes* enrichment broth. This would lead to a possible 1-day multiple contamination detection and identification of the most common *L. monocytogenes* CC. The first results are encouraging and will be the subject of a future publication.

## Conclusion

The results of this multiplex real-time PCR were reproducible among the 15 participating laboratories with high concordance values for molecular serotyping (100%) and CC identification (90.8%–100%). These results confirm the applicability of the method in other laboratories. The method was updated according to the improvements suggested by the validation trial (19).

## ACKNOWLEDGMENTS

We thank the staff of the European Union Reference Laboratory for *L. monocytogenes* for their contribution to the achievement of this project.

This work was conducted as part of the work program of the European Union Reference Laboratory (EURL) for *L. monocytogenes* (2023–2024). This work was co-funded by the European Union.

## AUTHOR AFFILIATIONS

[1]Salmonella and Listeria Unit, Laboratory for Food Safety, ANSES, European Union Reference Laboratory for Listeria monocytogenes, University of Paris-Est, Maisons-Alfort, France

[2]State Veterinary Institute, Jihlava, Czech Republic

[3]Unité Expertise analytique Laitière, Département Microbiologie Laitière, ACTALIA, La Roche sur Foron, France

[4]Food Safety Unit, ACTALIA, Saint-Lô, Normandy, France

[5]ADRIA Food Technical Institute, Quimper, France

[6]Laboratory for Food Safety, Bacteriology and Parasitology of Fishery and Aquaculture Products Unit, ANSES, Boulogne sur Mer, France

[7]Department of Fresh and Processed Meat, IFIP–The French Pig and Pork Institute, Pacé, France

[8]German Federal Institute for Risk Assessment, Berlin, Germany

[9]Microbiological National Reference Laboratory, National Food Chain Safety Office, Food Chain Safety Laboratory Directorate, Budapest, Hungary

[10]National Reference Labratory for Listeria monocytogenes, Istituto Zooprofilattico Sperimentale dell'Abruzzo e del Molise G. Caporale, Teramo, Abruzzo, Italy

[11]Norwegian Veterinary Institute, Ås, Norway

[12]National Institute for Agricultural and Veterinary Research (INIAV), Vila do Conde, Portugal

[13]Institute of Hygiene and Veterinary Public Health, Bucharest, Romania

[14]Swedish Food Agency, Uppsala, Uppsala County, Sweden

[15]Food Microbial Systems, Agroscope, Bern, Canton of Bern, Switzerland

[16]National Institute for Public Health and the Environment (RIVM), Bilthoven, the Netherlands

## AUTHOR ORCIDs

Karine Capitaine http://orcid.org/0009-0001-3431-8522
Thomas Brauge https://orcid.org/0000-0002-7649-3919
Marina Torresi http://orcid.org/0000-0002-4684-7886
Benjamin Félix https://orcid.org/0000-0002-8658-8874

## FUNDING

| Funder | Grant(s) | Author(s) |
| --- | --- | --- |
| European Commission | | Karine Capitaine |
| European Commission | | Sandrine Te |
| European Commission | | Adrien Asséré |
| European Commission | | Benjamin Félix |

## AUTHOR CONTRIBUTIONS

Karine Capitaine, Data curation, Formal analysis, Methodology, Project administration, Validation, Writing – original draft | Sandrine Te, Conceptualization, Methodology | Adrien Asséré, funding acquisition, Project administration | Valerie Michel, Formal

analysis, Writing – review and editing | Pauline Sabrou, Formal analysis, Writing – review and editing | Erwan Bourdonnais, Formal analysis, Writing – review and editing | Guillaume Gillot, Formal analysis, Writing – review and editing | Nassim Mouhali, Formal analysis, Writing – review and editing | Thomas Brauge, Formal analysis, Writing – review and editing | Cécile Dumaire, Formal analysis, Writing – review and editing | Carole Feurer, Formal analysis, Writing – review and editing | Baptiste Houry, Formal analysis, Writing – review and editing | Stefanie Lueth, Formal analysis, Writing – review and editing | Zsuzsanna Sréterné Lancz, Formal analysis, Writing – review and editing | Gabriella Centorotola, Formal analysis, Writing – review and editing | Fabrizia Guidi, Formal analysis, Writing – review and editing | Marina Torresi, Formal analysis, Writing – review and editing | Tone Mathisen Fagereng, Formal analysis, Writing – review and editing | Taran Skjerdal, Formal analysis, Writing – review and editing | Hugo Guedes, Formal analysis, Writing – review and editing | Gonçalo Nieto Almeida, Formal analysis, Writing – review and editing | Laurentiu Mihai Ciupescu, Formal analysis, Writing – review and editing | Paula Ågren, Formal analysis, Writing – review and editing | Monica Ricão, Formal analysis, Writing – review and editing | Elisabet Marti, Formal analysis, Writing – review and editing | Wilma Jacobs-Reitsma, Formal analysis, Writing – review and editing | Angela van Hoek, Formal analysis, Writing – review and editing | Benjamin Félix, Conceptualization, Methodology, supervision, Validation, Writing – original draft, Writing – review and editing.

## ADDITIONAL FILES

The following material is available online.

## Supplemental Material

**Table S1 (Spectrum00116-25-s0001.xlsx).** Description of the conditions applied by the participating laboratories.

## Open Peer Review

**PEER REVIEW HISTORY (review-history.pdf).** An accounting of the reviewer comments and feedback.

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
