## [Reviewer comments · Microbiology Spectrum]

Microbiology Spectrum

Inter laboratory validation trial report on multiplex real-time PCR method for molecular serotyping and identification of the 30 major Clonal Complexes of *Listeria monocytogenes* circulating in food in Europe

Karine Capitaine, Sandrine Te, Adrien Asséré, Hana Plodková, Valerie Michel, Pauline Sabrou, Erwan Bourdonnais, Guillaume Gillot, Nassim Mouhali, Thomas Brauge, Cécile Dumaire, Carole Feurer, Baptiste Houry, Stefanie Lueth, Zsuzsanna Sréterné Lancz, Gabriella Centorotola, Fabrizia Guidi, Marina Torresi, Tone Mathisen Fagereng, Taran Skjerdal, Hugo Guedes, Gonçalo Nieto Almeida, Laurentiu Mihai Ciupescu, Paula Ågren, Monica Ricão, Elisabet Marti, Wilma Jacobs-Reitsma, Angela van Hoek, and Benjamin Félix

Corresponding Author(s): Karine Capitaine, Anses

Review Timeline:

Submission Date:	January 23, 2025
Editorial Decision:	March 5, 2025
Revision Received:	April 22, 2025
Accepted:	April 23, 2025

Editor: Luca Cocolin

Reviewer(s): Disclosure of reviewer identity is with reference to reviewer comments included in decision letter(s). The following individuals involved in review of your submission have agreed to reveal their identity: Giovanna Franciosa (Reviewer #1)

Transaction Report:

DOI: <https://doi.org/10.1128/spectrum.00116-25>

Re: Spectrum00116-25 (Inter laboratory validation trial report on multiplex real-time PCR method for molecular serotyping and identification of the 30 major Clonal Complexes of *Listeria monocytogenes* circulating in food in Europe)

Dear Mx. Karine Capitaine:

Thank you for the privilege of reviewing your work. Below you will find my comments, instructions from the Spectrum editorial office, and the reviewer comments.

Revision Guidelines

Sincerely,
Luca Cocolin
Editor
Microbiology Spectrum

Reviewer #1 (Comments for the Author):

Please refer to the attached file.

Reviewer #2 (Comments for the Author):

Abstract:

- change "despite of DNA extraction" to "irrespective of DNA..."
- change "thermocyclers" to "thermocycler"

Introduction:

- first sentence something missing...listeriosis is a foodborne pathogen as far as humans are concerned and not technically a zoonotic issue? Clarify sentence.
- change "which is time consuming," to "which can be time consuming,"
- change "NRLs are in charge of epidemiological investigation in" to "NRLs are responsible for epidemiological investigations in..."
- change "This method splits" to "This method divides..."

Materials and methods:

- "the imposed to the participants." - please edit this to "were requested by each participant to..." to clarify what was being asked (i.e., the molecular serotyping)

Results:

- change "assigned the 20 strains to..." to "correctly identified the 20 strains to..."
- change to "For lab 7, the second attempt partially solved the failures. They..."

Discussion and perspectives:

- change to "and contamination, which were solved through..."
- change/edit last paragraph in this section. Is there any actual data that combined colonies from different CCs, for example, and the assay applied successfully? Consider rewriting to "A short-term, useful application of this assay is to test enrichment broths. This would lead to a possible one-day contamination and CC identification system for *Listeria* species."

This study presents the validation of twelve multiplex real-time PCR assays for identifying *L. monocytogenes* major molecular serotypes and clonal complexes through an interlaboratory study. The research is well-structured, and the results are robust.

My main concern relates to the reproducibility of the tests. In interlaboratory studies, reproducibility is a key factor and should be clearly defined and accurately calculated—typically by analyzing the variation in Ct values across different laboratories using identical samples. However, the manuscript does not report these reproducibility values, which should be included and discussed.

Below are some minor comments for the authors' consideration:

1. Page 2, "Importance" paragraph – The new real-time PCR method is proposed as a standard not only for rapid clonal complex identification but also for serotype identification. Please include this information.
2. Page 3, Introduction – I suggest to detail the four molecular groups identified through molecular serotyping. Also specify how many clonal complexes can be classified using MLST.
3. Page 4, Materials and Methods – The text states that the panel of 20 strains was optimized to reduce the number of DNA extractions required. However, this reasoning seems weak, as presumably DNA was extracted from all 98 strains for the validation of the remaining multiplex PCR assays. I recommend removing this sentence.
4. Table 1 – The table is unclear. The text states that only 20 strains were subjected to molecular serotyping; this should be clarified.
5. Figure 1 – Please specify what the PCR numbers (22, 78, 110, etc.) represent for each panel, given that 12 multiplex PCR assays were assessed.

Reviewer 1

My main concern relates to the reproducibility of the tests. In interlaboratory studies, reproducibility is a key factor and should be clearly defined and accurately calculated—typically by analyzing the variation in Ct values across different laboratories using identical samples. However, the manuscript does not report these reproducibility values, which should be included and discussed.

A statistical analysis with specificity, sensitivity and accuracy calculation was added to evaluate the performance of the molecular serotyping and CC identification, for each PCR, across different laboratories using identical samples. Table 3 with the results of calculation was added to the manuscript.

Line 158-170: Description of specificity, sensitivity and accuracy calculation in the material and methods paragraph.

Line 174-177: To highlight reproducibility results, a dedicated section was added dealing with real-time PCR condition variability in order to illustrate cause of variability.

Line 211-214: Calculation of the specificity, sensitivity and accuracy implied to present some false positive results for PCR CC21 and CC101. These FP had no impact as they relied on plca PCR results.

Line 222-234: A section dedicated to statistical analysis was added in the results part.

Line 242-244, 255-257, 262: A section dedicated to statistical analysis was added in the discussion part.

1. Page 2, "Importance" paragraph – The new real-time PCR method is proposed as a standard not only for rapid clonal complex identification but also for serotype identification. Please include this information.

Line 68: Precision provided

2. Page 3, Introduction – I suggest to detail the four molecular groups identified through molecular serotyping. Also specify how many clonal complexes can be classified using MLST.

Line 88: The four molecular groups were detailed

Line 95-97: The number of CC was specified up to the date of the manuscript revision.

3. Page 4, Materials and Methods – The text states that the panel of 20 strains was optimized to reduce the number of DNA extractions required. However, this reasoning seems weak, as presumably DNA was extracted from all 98 strains for the validation of the remaining multiplex PCR assays. I recommend removing this sentence.

Line 133-134: The sentence was removed

4. Table 1 – The table is unclear. The text states that only 20 strains were subjected to molecular serotyping; this should be clarified.

A clear separation was made between “strain requested by each participant to be used” and “strain chosen by the participants” to clarify the diversity of the strains chosen by the participants.

5. Figure 1 – Please specify what the PCR numbers (22, 78, 110, etc.) represent for each panel, given that 12 multiplex PCR assays were assessed.

Line 117: Precision provided on the different panels, connected to the table which was clarified

Figure 1 The PCR numbers (22, 78, 110) show the subtotal number of PCR performed, panel by panel. Symbols "+" and "=" were added to clarify the subtotal calculation. Global number of PCR performed was added in the bottom of the figure in a frame to make subtotal clearer. Out from reviewer comment, we realized that the 9 strains selected by the participants for molecular serotyping PCR assessment were not reported in the figure. It was corrected and modified in the new version of the figure

Reviewer 2

Abstract:

1. change "despite of DNA extraction" to "irrespective of DNA..."

Line 56: change done

2. change "thermocyclers" to "thermocycler"

Line 57: Reviewer 2 (2) change done

Introduction:

3. first sentence something missing...listeriosis is a foodborne pathogen as far as humans are concerned and not technically a zoonotic issue? Clarify sentence.

Line 76-78: The sentence was completed by a second that details the way of transmission to human

4. change "which is time consuming," to "which can be time consuming,"

Line 85: Change done

5. change "NRLs are in charge of epidemiological investigation in" to "NRLs are responsible for epidemiological investigations in..."

Line 100: Change done

6. change "This method splits" to "This method divides..."

Line 103 : Change done

Materials and methods:

7. "the imposed to the participants." - please edit this to "were requested by each participant to...." to clarify what was being asked (i.e., the molecular serotyping)

Line 129-130: The sentence was replaced

Results:

8. change "assigned the 20 strains to..." to "correctly identified the 20 strains to..."

Line 179: change done

9. change to "For lab 7, the second attempt partially solved the failures. They..."

Line 201: change done

Discussion and perspectives:

10. change to "and contamination, which were solved through..."

Line 241: Change done

11. change/edit last paragraph in this section. Is there any actual data that combined colonies from different CCs, for example, and the assay applied successfully? Consider rewriting to "A short-term, useful application of this assay is to test enrichment broths. This would lead to a possible one-day contamination and CC identification system for *Listeria* species."

Line 272-276: Clarification of the paragraph

Re: Spectrum00116-25R1 (**Inter laboratory validation trial report on multiplex real-time PCR method for molecular serotyping and identification of the 30 major Clonal Complexes of *Listeria monocytogenes* circulating in food in Europe**)

Dear Mx. Karine Capitaine:

Your manuscript has been accepted, and I am forwarding it to the ASM production staff for publication. Your paper will first be checked to make sure all elements meet the technical requirements. ASM staff will contact you if anything needs to be revised before copyediting and production can begin. Otherwise, you will be notified when your proofs are ready to be viewed.

Sincerely,
Luca Cocolin
Editor
Microbiology Spectrum